# Incidental Focal Spleen Lesions: Integrated Imaging and Pattern Recognition Approach to the Differential Diagnosis

**DOI:** 10.3390/diagnostics13152536

**Published:** 2023-07-30

**Authors:** Antonio Corvino, Vincenza Granata, Domenico Tafuri, Giulio Cocco, Orlando Catalano

**Affiliations:** 1Movement Sciences and Wellbeing Department, University of Naples “Parthenope”, Via Medina 40, I-80133 Naples, Italy; 2Division of Radiology, Istituto Nazionale Tumori IRCCS Fondazione Pascale, I-80131 Naples, Italy; 3Department of Neuroscience, Imaging and Clinical Sciences, University “G. d’Annunzio”, I-66100 Chieti, Italy; 4Radiology Unit, Varelli Diagnostic Institute, I-80126 Naples, Italy

**Keywords:** spleen, incidental findings, ultrasound (US), computed tomography (CT), magnetic resonance imaging (MRI), Positron Emission Tomography (PET)

## Abstract

Spleen lesions and pseudolesions, detected incidentally in imaging, are not uncommon and may require further work-up. The imaging appearance of focal splenic lesions (FSLs) may not be pathognomonic, because of considerably overlapping features. Consequently, all imaging techniques lack specificity to fully characterize FSLs. Clinical correlation is mandatory, so as, first of all, to categorize the patient as having or not having a history of solid or hematologic malignancy. Nowadays, many patients have old imaging studies available for comparison and, consequently, it is important to understand if the lesion was previously present or not, and if the size is the same or has changed. In the absence of comparison studies, and with a lack of imaging features of benignity, further investigation may be necessary, using PET, biopsy, or short-term follow-up. Some algorithms have been proposed to manage incidental FSLs; however, none of these strategies has been validated by prospective studies to date. In this review we illustrate the topic of incidental FSLs and we analyze a number of published algorithms.

## 1. Introduction

Spleen incidentalomas are unexpected abnormalities, detected within the spleen by imaging modalities and not related to the presenting illness of the patient. Although less common than other incidentalomas, such as those in the liver or kidney, occasionally-detected spleen abnormalities in asymptomatic subjects are nowadays encountered with an increased frequency [1,2,3]. This is due, on one hand, to the widespread use, and overuse, of abdominal or whole-body imaging examinations, and, on the other hand, to the increased sensitivity of the current imaging modalities, including X-rays, ultrasound (US), computed tomography (CT), and magnetic resonance imaging (MRI) [2,4,5,6,7,8].

Focal splenic lesions (FSLs) are detected in 0.1–0.6% of US examinations [9]. In a study of 3113 patients at a level I trauma center, Ekeh and colleagues reported on the prevalence of incidentally detected lesions on contrast-enhanced CT [10]. All lesions were benign. The prevalence of splenic granulomas was 1.4%, while both splenic cysts and hemangiomas were found in less than 0.08% of cases. Another study, at a level I trauma center, reported incidental splenic findings on contrast-enhanced CT in 1% of the cases, and all FSLs were regarded as benign [11]. In a retrospective study, evaluating the incidental findings in 876 patients with acute appendicitis evaluated with contrast-enhanced CT, 45% of the subjects had at least one incidental finding, including two splenic cysts, four splenic calcifications, and three splenic granulomas [12]. Incidental finding of an FSL on positron emission tomography (PET) is a rare, but possible, event.

In this review article we discuss the topic of splenic incidental lesions, illustrate the key features of splenic lesions and pseudolesions, and highlight findings suggestive of malignancy. Finally, we critically analyze a number of published management algorithms for spleen incidentalomas.

## 2. Incidental Spleen Abnormalities

### 2.1. Anatomical Variations

The normal spleen shows extensive variations in terms of size, shape, and contour. These changes can be congenital or acquired, due to previous trauma, infarcts, infection, surgery, etc. Natural incisions of the splenic surface (clefs) are reported with an autopsy prevalence of 40–98% and are especially frequent at the superior border of the organ [13]. Deep clefs may mimic spleen lacerations, in the setting of trauma.

In children, particularly if imaged with high-resolution transducers, the spleen may show a slightly heterogeneous echotexture, the so-called reticulo–nodular patterns, with diffuse millimetric hypo-echoic images representing white-pulp lymphoid follicles. This pattern should not be interpreted as an abnormality [14] (Figure 1).

### 2.2. Enhancement-Related Pseudolesions

The spleen can appear as inhomogeneous during the arterial phase of a contrast-enhanced US (CEUS), CT, or MRI, study [15,16]. This behavior, known as the zebra pattern, is due to the complex structure of the spleen, with different flow speeds between the white and the red pulps. This pitfall, that can be diffuse, but can also be focal, is clarified by the homogeneous appearance of the organ during the venous and late phases. However, sometimes, the incorrect interpretation of these pseudolesions may lead to further investigation.

### 2.3. Calcification

Spleen calcifications can be solitary or multiple. Causes include healed granulomatous disease (histoplasmosis, Pneumocystis carinii, etc.), Gamna–Gandy bodies, sickle cell disease, healed infarcts, hydatid disease, healed abscess, cysts, inflammatory pseudotumor, hamartomas, hemangiomas, lymphangioma, treated lymphoma, and metastases [9,17] (Figure 2). Splenic calcifications not associated to a mass, particularly if multiple and punctate (“starry sky” pattern), do not require further work-up. Calcifications may be seen with benign and malignant splenic masses [9]. When a lesion shows thick wall calcification (>5 mm), it can, however, be confidently diagnosed as benign [18].

### 2.4. Cystic Lesions

Spleen cysts include parasitic (Echinococcus granulosus), congenital (2.5% of the cases), pseudocysts, and neoplastic cysts (lymphangioma, hemangioma, metastasis from ovarian carcinoma) [16,17]. A solitary splenic cyst is almost always benign [19]. Size may vary considerably and the cyst can be unilocular or multilocular [20]. The content is anechoic at US, hypo-attenuating at CT (0–20 HUs), hypo-intense in T1-W MR images, and markedly hyperattenuating on T2-W MR images [21]. Septations and trabeculations are uncommon in post-traumatic cysts, but common in congenital cysts [22]. The wall can be imperceptible; particularly in the case of true cysts, or when thick and calcified, especially in the case of pseudocysts [19]. In any case, contrast enhancement is completely missing. Cystic lesions with a dense content may mimic a solid lesion, particularly on US and on unenhanced CT (Figure 3). At the same time, apparently solid lesions may lack any contrast enhancement on US, CT, or MRI, turning out to be cystic [23].

The spleen is the third most common site for hydatid cysts, and, therefore, in some geographical regions, Echinococcus granulosus is one of the most common causes of spleen cystic lesion [9]. Characteristic US appearances are usually found, and contrast studies constantly shows the lack of contrast enhancement [9].

Cystic content may be heterogeneous, particularly in the case of pseudocysts. Hemorrhage may complicate splenic cysts. On CT, these lesions will be hyperattenuating, sometimes exhibiting fluid–fluid levels. On MRI, hemorrhagic cysts usually demonstrate high signals on T1W scans and may show internal levels. There is no contrast enhancement at CT or MRI [24].

### 2.5. Hematoma and Abscess

The probability of incidentally encountering a spleen parenchyma hematoma or abscess in an asymptomatic patient is quite low. Hematomas are almost always due to recent blunt traumas, while abscesses are usually symptomatic [19,25,26,27]. Abscesses (pyogenic, more commonly Gram-negative bacteria, and fungal, mostly Candida species, or tuberculous) are usually identified in specific scenarios, such as subjects with immunosuppression (organ transplantation), AIDS, hematological malignancies, bacterial endocarditis [20]. Concomitant liver, renal, and/or nodal involvement is usually present, particularly in the case of fungal infection. Number and size may vary. At US, abscesses are round, hypo-echoic, with increased dorsal enhancement [20]. Margins can be well- or ill-defined. Abscesses may contain echogenic components, due to debris, or hyperechoic foci, due to air bubbles. Doppler techniques and CEUS may demonstrate hypervascularity within the thick wall but always show absence of internal flows [15,28]. Candida infections show a “bulls-eye” or a “wheel within a wheel” pattern, consisting of different layers of necrosis, inflammation, and fibrosis [28]. On CT, abscesses are hypo-attenuating and hypo-enhancing. A peripheral ring enhancement may be present [20,29].

### 2.6. Infarcts

Spleen infarctions are usually caused by thromboembolic conditions, mostly originating from atrial fibrillation, vasculitis, or splenic vein thrombosis. alternatively, a splenic artery infiltration, such as in the case of pancreatic tail carcinoma, may determine an infarction. During transcatheter embolization procedures or abdominal surgery, an iatrogenic origin is also possible. Although larger spleen infarctions are usually painful, it is possible to encounter these changes incidentally, for example during cancer patient staging or follow-up. This happens particularly in the case of smaller and/or polar infarctions. Splenic infarcts usually appear as hypo-echoic/hypo-attenuating, wedge-shaped defects, extending to the capsule [20]. These lesions do not exhibit mass effect, Doppler signal, or contrast enhancement. However, atypical appearances are possible, such as for round-shaped infarcts mimicking solid lesions (Figure 3).

### 2.7. Solid Lesions

Splenic hemangioma, single or multiple, is reported with an autopsy prevalence of up to 14% [30]. The size is usually less than 2 cm, although larger hemangiomas are possible [17,21]. At US, hemangiomas may appear as hyper-echoic or, less frequently, as hypo-echoic. Smaller hemangiomas are usually homogeneous, while larger ones may show a complex solid and cystic appearance [28]. Punctate or curvilinear peripheral calcification is possible [21]. On Doppler imaging there may be peripheral vascularity or absent vascularity [28]. At unenhanced CT, the lesion is iso-attenuating or, more frequently, hypo-attenuating [29]. Hemangiomas are hypo- to iso-intense on T1-W and hyper-intense on T2-W MRI scans. These lesions have variable contrast behavior at CT and MRI, from subtle to intense and diffuse enhancement, to centripetal fill-in [18,24] (Figure 4). Hemangiomas with atypical appearance on contrast imaging may mimic metastasis. Due to the differences in timing during the administration of contrast material, hemangiomas may appear slightly larger or smaller on follow-up images [16]. This pseudogrowth should not be misinterpreted as through growth.

Lymphangiomas are found more frequently in children and may be cystic (uniloculated or, more frequently, multiloculated) or, less commonly, capillary or cavernous [16,17]. They are usually located close to the spleen capsule and range in size between a few millimeters and several centimeters. At imaging assessment, these lesions appear thin walled and sharply marginated, without any significant contrast enhancement. Vascularity along the walls in Doppler imaging is helpful for the diagnosis [28]. Intervening fibrous stroma can appear as septa, with calcifications or progressive enhancement [19]. Some components can appear as echoic at US or as hyper-intense on T1-W MR images because of the proteinaceous or hemorrhagic contents [18]. Fat areas can also be present. Lymphangiomas can be confused with metastases, especially if multiple, and detected in a cancer patient. However, lymphangiomas have low signal intensity on T1-W images, high signal intensity on T2-W images, and do not enhance [18].

Hamartomas have an incidence on autopsy series ranging between 0.024% and 0.13%, without any gender or age predilection, although size is usually larger in females [17,19]. These lesions are usually solitary, large (mean, 5 cm), and well-circumscribed [21,22]. US shows a hypo-echoic or iso-echoic appearance, slightly heterogeneous, with significant flows (Figure 5). Hamartomas are typically hyper-intense on T2-W images. Calcifications and cystic changes can be present [19]. Contrast imaging at US, CT, and MR show intense vascularization during the arterial phase and iso- or hyper-enhancement in the venous and late phases [18,24] (Figure 6). Contrast enhancement is usually homogeneous in smaller lesions and heterogeneous, but prolonged, in larger ones [21]. The strong contrast uptake of the spleen during the arterial phase may, however, masquerade the hamartoma, and it may, consequently, be missed.

Lymphoma represents the most common malignant tumor of the spleen. However, isolated primary lymphoma of the spleen is quite uncommon, representing less than 2% of all lymphomas [17]. Lymphoma can present as splenomegaly without focal lesions, multiple small or large lesions, or a solitary lesion. Lymphomatous nodules usually appear as homogeneous and hypo-vascular [19]. Margins can be well- or ill-defined. These lesions are almost always hypo-echoic at US, sometimes near anechoic. Scattered penetrating vascularity may be seen in Doppler imaging [19]. The MR appearance is usually iso-intense in T1-W and T2-W images [21]. Contrast enhancement is usually low and peripheral (Figure 7). Hilar lymphadenopathies are frequently, but not constantly, present [9].

The spleen represents an uncommon site of solid tumor metastasis. In a series, spleen metastases were found in 0.6% of autopsies, with 20% of the cases consisting of microscopic lesions [31]. In another study 1% of cancer patients developing metastasis had a splenic involvement [32]. Metastatic lesions to the spleen can be hematogenous or may develop as capsular deposits in subjects with peritoneal carcinosis. Primary tumors that most frequently cause synchronous or metachronous splenic metastases include breast carcinoma, lung carcinoma, ovarian carcinoma, stomach carcinoma, and melanoma. Metastases to the spleen usually occur in subjects with a multi-visceral, already-known cancer, while a solitary splenic metastasis represents quite a rare occurrence [9,31,32] (Figure 8). However, it must be kept in mind that not all FSLs associated with extra-splenic malignancy are metastases. These lesions can be single or, more commonly, multiple, and show limited vascularization [22]. At US, half the cases are hypo-echoic and half are heterogenous, cystic, or even hyper-echoic [28]. During MRI, metastases have low signal intensity on T1-W images and relatively high signal intensity on T2-W images. Melanoma metastasis may be hyper-intense on T1-W scans. Spleen metastases are often poorly marginated and heterogeneous, especially on T2-W images and in post-contrast US, CT, and MR scans, where the contrast uptake is usually peripheral [18,21].

Primary splenic malignancies are quite rare, and affected patients are usually symptomatic at time of presentation [9]. Angiosarcoma is the most frequent one, typically present in patients more than 40 years of age, with no gender predilection. It can be single or multiple and may show a contrast enhancement resembling that of hemangioma. This tumor can appear as an ill-defined, heterogenous mass with intensely enhanced components and focal areas of non-enhancement [9,18]. Area of hemorrhage and hemosiderin deposits may appear as spontaneously hyper-attenuating foci at unenhanced CT and as portions of variable signal intensity on T1-W and T2-W MR images [21]. ADC levels are low [21].

## 3. Risk of Malignancy and Pattern Recognition

In subjects with no history of cancer, the probability that an incidentally detected spleen lesion is due to a primary tumor or to a metastasis is very low. The current literature for adults estimates the incidence of non-lymphomatous metastasis to the spleen at 1% [20]. In a series of patients younger than 30 years old, and with an incidental spleen lesion, only 3% of the subjects without a known history of malignancy resulted in a malignant finding [33]. On the other hand, 87% of the lesions identified in patients undergoing cancer follow-up were malignant, although it is debatable if this occurrence represents a truly incidental finding or not.

In most of the cancer patients undergoing staging or follow-up, a splenic lesion is usually benign, especially if single [25]. Since solitary splenic metastases are quite rare, a splenic lesion detected in a cancer patient with no concomitant evidence of metastases in other organs, has a very high likelihood of benignity. However, clearly, there is always the need to be certain of this benignity since the detection of a metastasis may alter the disease stage and dramatically impact patient management. Similarly, most splenic lymphoma lesions are seen in patients with systemic disease and concomitant evidence of lymphoma is found elsewhere [25]. However, especially at abdominal US, it is possible that no other location is actually visible and that the spleen abnormality is apparently solitary at the moment of the examination.

The work-up of FSLs is difficult, due to the large number of different entities, the low rate of relevant primary lesions, the high rate of splenic lesions resulting from extra-splenic diseases with focal splenic involvement, the high percentage of benign lesions in asymptomatic patients, and the low accuracy for characterization of all imaging methods, compared with that for other organs [34]. Some questions can help in clarifying the diagnosis. Is it truly an incidental lesion in an asymptomatic patient? Does the patient have fever or pain that could be related to the spleen? Does the patient have a known underlying malignancy or any other signs of metastatic disease? Is the patient immunocompromised? Does the patient suffer from sarcoidosis, tuberculosis, or sickle cell disease? Is there a history of trauma or endocarditis? Some other questions concern the imaging findings. Is the spleen enlarged or not? Is there an isolated FSL or are there multiple FSLs? What is the appearance in unenhanced imaging and at the various phases of contrast-enhanced imaging? Is there any other associated abdominal condition, such as liver lesions or adenopathy? When reporting an imaging study, any incidental spleen abnormality must be indicated, since it may require an appropriate characterization, to rule out, first of all, anything of a malignant nature. If necessary, further imaging may be based on additional work-up with other imaging modalities, or on demonstrating the presence or absence of changes at patient follow-up. The radiologist should be aware of a number of confusing lesions, such as littoral-cell angioma, inflammatory pseudotumors, peliosis, Gaucher nodules, tuberculomas, and sarcoidosis lesions. Although uncommon, these abnormalities may create difficulties in terms of differential diagnosis, mimicking a malignant lesion or, at the least, showing a non-specific appearance, requiring further assessment.

Lesions size matters. Heller and coworkers stated that in patients with a history of a primary neoplasm with a tendency to metastasize to the spleen, an incidentally discovered splenic lesion larger than 1 cm should be suspected as being metastasis while, even in the face of a known neoplasm, a very small lesion is likely to be benign [18].

Various abnormalities may result in complex splenic cysts, including hemorrhagic cyst, hematoma, lymphangioma, infarction, metastases, and primary malignancy. MRI is especially useful for cystic SFL diagnosis.

FSLs with benign imaging characteristics (anechoic at US, homogeneous, low-attenuation, smooth margins, and no contrast enhancement at CT or MRI) do not require further imaging investigation or follow-up [24]. If the imaging features are not diagnostic, but there is stability, based on prior examinations, no work-up is indicated. In the remaining cases further investigation or follow-up is usually necessary.

In a retrospective study, Abrishami et al. evaluated the features that could aid in the differentiation of benign and malignant FSLs [35]. Benign lesions were more likely to be cystic (22% vs. 3%), homogenous (60% vs. 30%) and to demonstrate well-defined borders (69% vs. 30%). Malignant lesions had significantly larger diameters (median size: 15 vs. 11 mm). During MRI restricted diffusion was not seen in any of the benign lesions; however, 50% of malignant lesions demonstrated restricted diffusion. Features such as lesion distribution, presence of calcification, splenomegaly and number of lesions were not significantly different between benign and malignant lesions.

Hypo-echoic lesions represent the greatest diagnostic challenge at US [9]. Echo-poor masses can be due to infarct, hemangioma, lymphangioma, hamartoma, sclerosing angiomatoid nodular transformation, peliosis, inflammatory pseudotumor, abscess, tuberculosis, sarcoidosis, lymphoma, multiple myeloma, sarcoma, or metastasis [9,28,36]. Some of these lesions are typically solitary, while some others are usually multiple.

In a retrospective CT study on splenectomy patients, common findings of malignant lesions included their being enhanced, mainly solid, having ill-defined margins, absence of splenomegaly, absence of the walls, absence of calcification, enlargement of lymph nodes, and the presence of underlying malignancy [37]. Heller and co-workers categorized the imaging features of FSLs at CT and MRI into benign, indeterminate and suspicious [18]. Lesions were diagnosed as benign in cases of cysts (imperceptible wall, near-water attenuation [<10 HUs], no contrast enhancement), hemangioma (discontinuous, peripheral, centripetal enhancement), benign (homogeneous, low attenuation [<20 HUs], no enhancement, smooth margins). Lesions were called indeterminate if they were heterogeneous, had intermediate attenuation (>20 HUs), enhancement, and smooth margins. Lesions were finally regarded as suspicious if they showed heterogeneity, enhancement, irregular margins, necrosis, parenchymal or vascular invasion, and substantial enlargement.

Ill-defined margins and hypo-vascularity on CEUS, contrast-enhanced CT, or contrast-enhanced MRI are predictors of malignancy [9]. In a retrospective study on the CT and MRI differentiation of FSLs in 79 pathologically proved cases, the combination of ill-defined margin and hypo-vascular enhancement for suggesting malignant lesions had 95% specificity and 90% accuracy [38]. A low signal intensity on the three minute delayed phase and a diffusion restriction were the two most reliable MRI findings for the differentiation of malignant from benign splenic lesions [39]. Mainenti et al. retrospectively correlated the contrast-enhanced PET/CT patterns of FSLs with the cytohistological reports [40]. A significant association was observed between focal hyperattenuating lesion, infarcts/cysts, and focal photopenia/diffuse uptake inferior to that of the liver and benignity. On the other hand, a significant association with malignancy was proved for splenomegaly without focal lesions, focal hypo-dense lesions, diffuse uptake superior, or equal, to that of the liver, and focal increased uptake.

Hypo-attenuating FSLs are frequently encountered in the portal–venous phase CT images of the abdomen. The majority of these findings represent benign lesions that require no further work-up. However, certain appearances, such as ill-defined lesion borders, presence of solid, contrast-enhancing components, and increased attenuation of the lesion, may be related to a potentially more relevant disease [16]. In patients with sickle cell disease, the presence of multiple small hypo-dense lesions is strongly suggestive of splenic infarctions [16]. Sarcoidosis can be suspected in young or middle-aged women with multiple, small (approximately 10 mm), hypo-echoic–hypo-attenuating lesions of the spleen and abdominal lymphadenopathy, most commonly at the porta hepatis and paraaortic region [17,20,41]. Spleen lesions are hypo-enhanced both in the arterial and in the venous phase [41]. Chest CT may be indicated to look for lung or nodal involvement. Multiple incidental complex nodules within the spleen of an immunocompromised patient should also raise the possibility of fungal (candida) infection [9].

Hyper-vascular lesions in the spleen include pseudoaneurysms, hemangiomas, hamartomas, sclerosing angiomatoid nodular transformation of the spleen, littoral cell angioma, angiosarcoma, pleomorphic sarcoma. Lymphoma and metastases are rarely hyper-enhancing, while sarcomas are quite uncommon [25,42,43].

CEUS has been found comparable to CT and MRI in differentiating FSLs [44,45]. Current US contrast media are blood-pool and their behaviors directly reflect the lesion perfusion, compared with the spleen parenchyma perfusion. During real-time assessment it is possible to evaluate the presence, intensity, and length of the lesion wash-in (enhancement), as well as the presence of a wash-out effect (detersion), being quick and marked or slow and moderate. Malignant FSLs typically manifest at CEUS imaging variable arterial enhancement and venous–phase washout. Benign lesions typically manifest no contrast enhancement or iso- or hyper-enhancement in the arterial phase with lack of washout in the venous phase. Benign lesions may sometime show some detersion but this behavior is usually slow and moderate, while quick and marked washout is an indicator of malignancy until the contrary is proved [15]. Hence, prolonged contrast enhancement is usually an indicator of benignity, while wash-out may be due to benign lesions or, most frequently, but not necessarily, to malignant lesions. An FSL that is constantly non-enhanced or iso-enhanced with adjacent splenic parenchyma in the venous phase is invariably benign, whereas an FSL manifesting venous–phase hypo-enhancement is predictive of malignancy in 87% of cases [9]. Intralesional vessels, inhomogeneous enhancement, necrotic, non-enhancing areas, and dotted enhancement are additional features suggesting malignancy [9,15].

FSLs showing an avid ^18^F-fluorodeoxyglucose (FDG) uptake at PET imaging are usually malignant. However, it should always be considered that benign lesions (sarcoidosis, inflammation, abscess) may also sometimes exhibit some kind of metabolic activity at PET imaging [21]. In patients with known oncologic pathology, a standardized uptake value threshold of 2.3 accurately differentiates between benign and malignant FSLs [46]. On the other side, PET may be false negative for tiny or necrotic malignant lesions and for non-avid primary tumors (thyroid, kidney, etc.), as well as for well-differentiated lymphomatous lesions [47].

## 4. Spleen Lesions Sampling

The fear of bleeding from such a vascularized organ with a very thin capsule has always mitigated against the performance of percutaneous biopsies of the spleen. When concomitant abnormalities exist in other organs, histological sampling is usually performed from non-splenic sites [9]. However, if no other abnormality is present and there is a persistent suspicion of malignancy, cytological sampling of the spleen may represent the only way to achieve a definitive diagnosis [9]. Splenic fine-needle biopsy is now being performed with increasing frequency, with a good success rate and an acceptable rate of major complications (2%) [48,49]. In a study of the biopsies of FSLs appearing as hypo-perfused in late-phase CEUS yielded a high rate of malignancy [34]. The most common indication for spleen biopsy is indeterminate lesion in a patient with a history of lymphoma or other malignant disease [25].

## 5. Published Algorithms

In 2005, Metser and co-workers built a PET-based algorithm for spleen masses detected at imaging [46]. Two groups of patients were retrospectively assessed: 68 patients with known malignancy and a focal lesion on PET or a solid mass on CT portions of the ^18^F-FDG PET/CT study; 20 patients with solid splenic masses on conventional imaging without known malignancy. The sensitivity and specificity, of PET/CT in differentiating benign from malignant solid splenic lesions in patients with and without malignant disease were 100% and 100% vs. 100% and 83%, respectively. Based on the debatable assumption that “conventional imaging often cannot differentiate benign from malignant solid splenic masses”, the authors created an algorithm. For patients with splenic mass on conventional imaging and a known malignancy, PET imaging was indicated, with further imaging follow-up suggested for non-avid lesions. For patients without a known malignancy, PET was indicated as an alternative to imaging follow-up, with PET suggested in the case of growth at follow-up. The main limitation of this algorithm is the high cost of a PET exam, considering the low probability of malignancy in subjects with no history of malignancy. Appling the suggested approach means, for example, submitting a patient with a non-specific, small, hypo-attenuating lesion of the spleen at contrast-enhanced CT, which is quite a frequent occurrence, usually due to hemangiomas/lymphangiomas, to PET/CT. Additionally, nothing is said on how to manage US incidentalomas, on the choice between follow-up or immediate investigation, and whether to use CT or MR imaging for further imaging.

In 2013, the American College of Radiology (ACR) published a white paper on the management of incidental spleen findings in CT and MRI [18]. Unfortunately, US was not considered, neither as a common source of incidental FSLs nor as an imaging modality to characterize (CEUS) or follow-up these lesions. The ACR algorithm starts from the imaging features. In the case of benign appearance, no further work-up or follow-up is required. In patients with non-diagnostic features, prior studies are then considered. Cases with prior imaging that have been stable for more than one year do not require follow-up, while cases with prior imaging and size change need evaluation. Cases without prior imaging require evaluation (PET vs. MRI vs. biopsy) if there is any cancer history and larger lesions (≥1 cm) or if there is no cancer history but suspicious imaging features. Instead, the recommendation was an MRI follow-up in six and twelve months for patients with no cancer history and indeterminate features and for patients with a cancer history but lesions smaller than 1 cm.

The retrospective study conducted by Dhyani and coworkers focused on solitary, non-simple, cyst lesions of the spleen detected in young patients (<30 years old) [33]. These lesions (26% malignant) had been detected incidentally in 53 subjects, using CT (51% of cases, most with a single portal-phase acquisition) but also US, MRI, or PET/CT. History of malignancy was the basis for the algorithm built in this study. The authors recommended MRI for imaging work-up of an incidental spleen lesion in patients with no history of malignancy, since this modality enabled a definitive diagnosis of most benign lesions. In the case of malignancy history (lymphoma in 64% of cases), the recommendation was instead for PET/CT, though this technique proved less helpful for characterization of isolated splenic lesions. Lesions with indeterminate MRI features could be followed-up with US or CT. These MRI-based recommendations seem to be reasonable, because of the concerns around the ionizing radiation exposure associated with CT. The cost of contrast-enhanced MRI to characterize benign, eventually irrelevant, lesions of the spleen should, however, always be considered. Assessment with CEUS or follow-up US could be a valuable alternative in our opinion, particularly for low-risk patients, when the small lesions are found incidentally at US.

In a review from Thut and colleagues [24] patient clinical status represented the primary categorization basis for patients with incidental FSLs. As a matter of fact, patients were triaged into asymptomatic, symptomatic with pain attributable to the spleen and symptomatic with systemic manifestations. For asymptomatic subjects (i.e., patients with truly incidental lesions), MRI was recommended for all complex cysts and solid lesions. This implies that CT should not be performed in subjects with incidental spleen abnormalities at US. If the lesion has signal and enhancement MRI characteristics of a hemangioma or hemorrhagic cyst, no follow-up imaging is recommended. Follow-up imaging in 6–12 months was recommended in the case of indeterminate MRI findings.

In the year 2021, based on a previous study by Ignee et al. [34], the World Federation for Ultrasound in Medicine and Biology (WFUMB) published a position paper on incidental FSL management. These guidelines suggested the CEUS-based triage for echo-poor FSLs detected at US [9]. Management criteria included symptoms (yes, no), past medical history (malignancy, inflammatory disease), distribution (solitary, multiple), echogenicity (no further assessment for echo-free lesions and 3-months follow-up with US for echo-rich lesions), contrast enhancement with or without washout, and stable size or progression during follow-up. Further diagnostic investigation (additional imaging, biopsy) was recommended for FSLs with low-level arterial enhancement and progressive late-phase wash-out at CEUS assessment [9].

In 2022, Kim et al. published an imaging-based algorithm [19]. Not taking into consideration whether the diagnosis was incidental or not, the FSLs were categorized into four groups: solitary cystic, solitary solid, multiple cystic, and multiple solid. For each of these categories, the key findings were listed. No information was provided, however, on the management of non-specific appearances and on the imaging modalities to employ for further investigation or follow-up. In the end, this was not properly an algorithm.

To our knowledge, there is no published article on texture analysis applied to the differential diagnosis of FSLs.

## 6. Conclusions

Spleen lesions are relatively uncommon and most are detected incidentally. Correlation with the context is essential. Clinical factors must be taken into account; most importantly, left upper quadrant pain, symptoms of infection, immune status, history of known malignancy, white blood cell count, associated imaging findings. Most incidentally detected spleen masses are benign. However, application of specific imaging criteria can help determine if a lesion should undergo further imaging or follow-up. The management of incidental FLLs may vary between single institutions and also among different countries, since an intensive use of expensive imaging modalities is not s possible everywhere. In underdeveloped countries, patients may not have imaging documentation from earlier conditions, and, due to financial conditions, further imaging techniques may not be possible. In this setting, follow-up may represent a more realistic choice than further investigation.

## Figures and Tables

**Figure 1 diagnostics-13-02536-f001:**
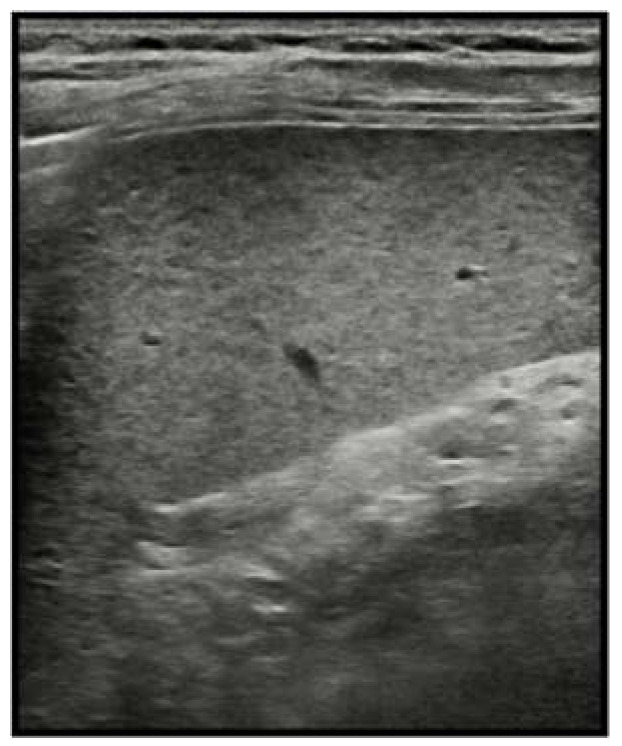
US scan of 8-year-old man examined with a 7 MHz frequency. Normal reticulo–nodular pattern of the splenic echotexture mimicking hypo-echoic micronodular infiltration.

**Figure 2 diagnostics-13-02536-f002:**
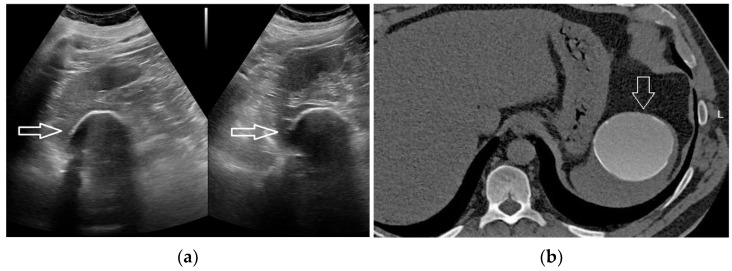
A 68-year-old man undergoing US because of a gallbladder stone. A large calcification with strong back-shadowing was detected within the spleen upper pole (**a**, arrows). Unenhanced CT scan (**b**, bone window setting) demonstrating a fully calcified cyst. The case was interpreted as splenic hydatid disease and no further imaging was carried out. The arrow indicates the calcified cyst.

**Figure 3 diagnostics-13-02536-f003:**
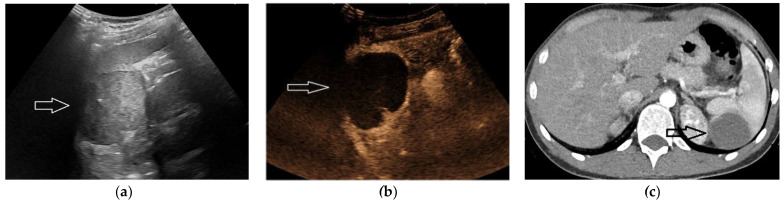
A 29-year-old man submitted to US screening because of blunt abdominal trauma. US scan detected a large, hypo-echoic mass within the upper pole of the spleen (**a**, arrow). Arterial-phase CEUS scan (**b**, arrow) demonstrating the fully avascular nature of the mass. Arterial-phase CT scan confirmed the finding (**c**, arrow). Retrospectively, the CT exam could have been avoided.

**Figure 4 diagnostics-13-02536-f004:**
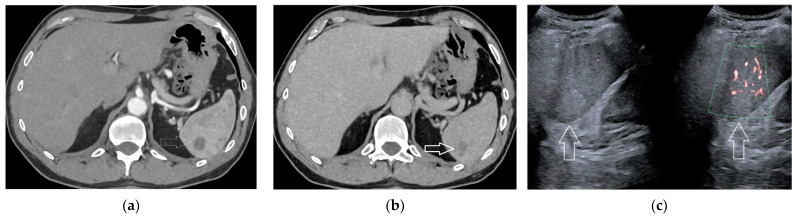
A 55-year-old woman undergoing contrast-enhanced CT because of lung lymphoma. Arterial-phase (**a**, arrow) and late-phase (**b**, arrow) CT scans demonstrating a hypo-perfused spleen nodule with partial, progressive fill-in. US was then performed (**c**, arrows), showing a hyperechoic, non-vascularized lesion. The finding was interpreted as a splenic hemangioma. A subsequent PET study (not shown) did not detect any spleen uptake and a 3-mo. US follow-up did not show any change.

**Figure 5 diagnostics-13-02536-f005:**
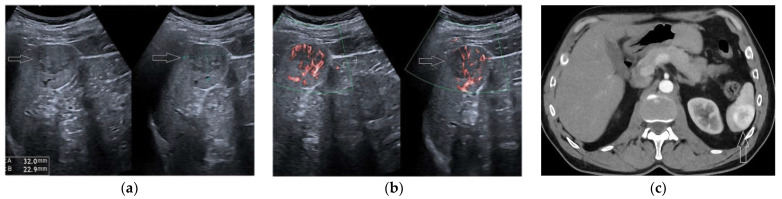
A 60-year-old woman submitted to abdominal US because of renal stone disease. US (**a**, arrows) incidentally detected a 32 × 23 mm, heterogeneously hypo-echoic mass at the spleen lower pole. The lesion showed a strong and diffuse vascularization (**b**, arrows). Subsequent arterial-phase CT scan (**c**, arrow), confirmed the hypervascularity of the mass.

**Figure 6 diagnostics-13-02536-f006:**
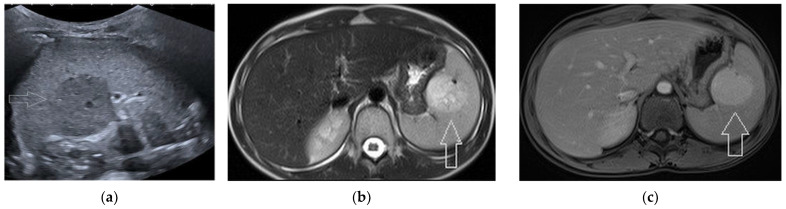
A 12-year-old female undergoing abdominal US because of post-viral pancreatitis. US scan (**a**, arrow) detected a 34 × 33 mm hypo-echoic splenic mass. Subsequent contrast-enhanced MRI demonstrated an hyperintense mass (**b**, T2-W scan, arrow) with persistent enhancement (**c**, portal-phase scan, arrow).

**Figure 7 diagnostics-13-02536-f007:**
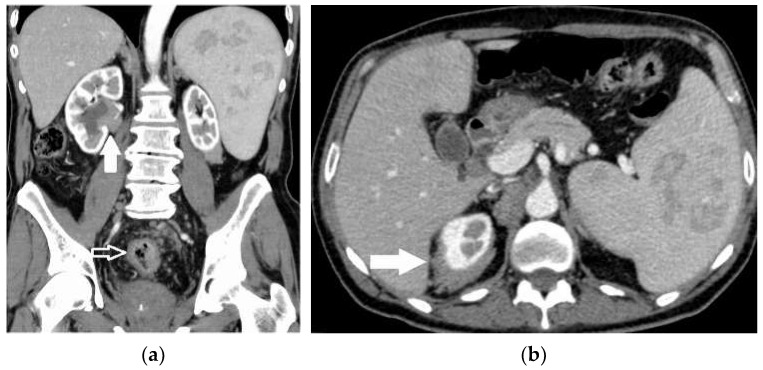
A 70-year-old man undergoing CT scanning because of colon cancer staging. Arterial-phase CT scans (**a**,**b**) showing the sigmoid colon mural thickening (empty arrows) but incidentally detecting a peri-ureteral lymphomatous spread on the right (full arrows) and an enlarged spleen with hypo-echoic nodules. The case was classified as synchronous colon carcinoma and non-Hodgkin lymphoma.

**Figure 8 diagnostics-13-02536-f008:**
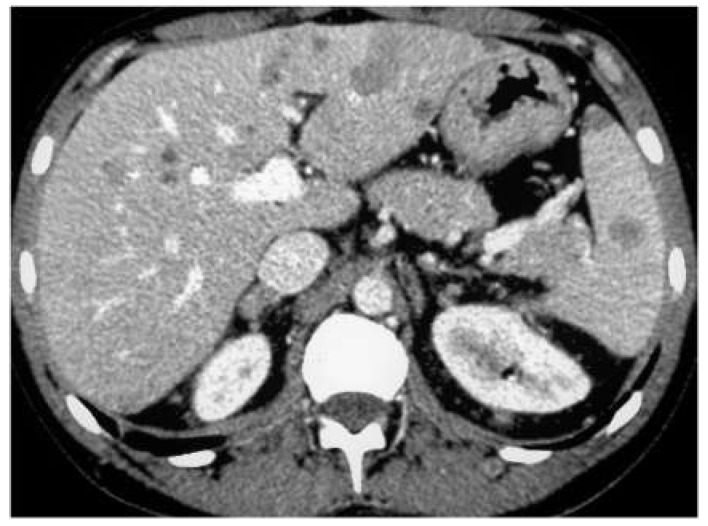
A 70-year-old man undergoing CT because of melanoma staging. Venous-phase CT scan showing multiple hypo-attenuating nodules within the liver and the spleen.

## Data Availability

The data presented in this study are available on request. These data are not publicly available due to restrictions regarding patients’ privacy.

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
