# Peer review of "Incidental Focal Spleen Lesions: Integrated Imaging and Pattern Recognition Approach to the Differential Diagnosis"

_diagnostics, 2023, doi:10.3390/diagnostics13152536_

Round 1

Reviewer 1 Report

The authors are conducting a thorough examination of the existing literature on the subject of Focal Splenic Lesions (FSLs). They emphasize the identification of pattern recognition and imaging characteristics that can be observed in FSLs. While certain aspects mentioned are widely recognized, it is worth highlighting particular patterns and approaches, particularly those related to the use of basic US and Contrast-Enhanced Ultrasound (CEUS) techniques. Although this article aims to provide a comprehensive review, it is essential for the authors to also explore the literature on Artificial Intelligence (AI)-based algorithms in Title: ok

Abstract: ok

Typo In line 32: “such as those the liver or kidney”

Typo in line 386- . In patients with non-diagnostic features, the availability of prior studies in then considered spleen imaging, given the current era of advancements in AI.

Author Response

MANUSCRIPT REVISION

REVIEWER #1

1)The authors are conducting a thorough examination of the existing literature on the subject of Focal Splenic Lesions (FSLs). They emphasize the identification of pattern recognition and imaging characteristics that can be observed in FSLs. While certain aspects mentioned are widely recognized, it is worth highlighting particular patterns and approaches, particularly those related to the use of basic US and Contrast-Enhanced Ultrasound (CEUS) techniques. Although this article aims to provide a comprehensive review, it is essential for the authors to also explore the literature on Artificial Intelligence (AI)-based algorithms in Title: ok

[rev1, comment1]

We have added more info on the CEUS findings for focal splenic lesions. As we wrote in the text, “To our knowledge, there is no published article on texture analysis applied to the differential diagnosis of FSLs.” We were unable to find any article dealing with AI-based algorithms for splenic lesions

2)Typo In line 32: “such as those the liver or kidney”

[rev1, comment2]

Thank you, the typo was correct. “in” was added  

3)Typo in line 386- . In patients with non-diagnostic features, the availability of prior studies in then considered spleen imaging, given the current era of advancements in AI.

[rev1, comment3]

Thank you, the typo was correct, from “in” to “is”  

Reviewer 2 Report

In this manuscript (review), authors discuss splenic incidental lesions, including key features of lesions and pseudo lesions, and findings suggestive of malignancy. The authors also analyze published management algorithms for spleen incidentalomas.

The author's conclusions are:

Spleen lesions are relatively uncommon, and most of them are detected incidentally. Correlation with the context is essential. Clinical factors must be considered, most importantly left upper quadrant pain, symptoms of infection, immune status, history of known malignancy, white blood cell count, and associated imaging findings. Most incidentally detected masses are benign. However, specific imaging criteria can help determine if a lesion should undergo further imaging or follow-up.

The topic of the review article is not sufficiently covered in the literature, so it is important for medical diagnostics.

Minor:

In underdeveloped countries, patients do not have imaging documentation in earlier conditions, and due to their financial condition, they cannot apply more imaging techniques. Therefore, the authors could rank imaging techniques according to the diagnostic importance, i.e., how the diagnostic significance changes of this imaging technique in diagnosing spleen lesions.

The authors could also schematically provide an algorithm (protocol) for the diagnosis of spleen lesions.

Author Response

MANUSCRIPT REVISION

REVIEWER #2

1)In underdeveloped countries, patients do not have imaging documentation in earlier conditions, and due to their financial condition, they cannot apply more imaging techniques. Therefore, the authors could rank imaging techniques according to the diagnostic importance, i.e., how the diagnostic significance changes of this imaging technique in diagnosing spleen lesions.

[rev2, comment1]

We have added the following sentences regarding underdeveloped countries:

“The management of incidental FLLs may vary between the single institutions and also among the different countries, since an intensive use of expensive imaging modalities is not poossible everywere. In underdeveloped countries, patients may not have imaging documentation in earlier conditions, and due to their financial condition, they cannot apply more imaging techniques. In this setting, follow-up may represent a more realistic choice than further investigation”.  

2)The authors could also schematically provide an algorithm (protocol) for the diagnosis of spleen lesions.

[rev2, comment2]

As illustrated in this manuscript, there are already several existing algorithms and all of them can be criticized from several points of view. It is really not over goal to provide on more algorithm, since this would only add further confusion. We would build a diagnostic algorithm solidly based on CEUS, given also the fact that in our country many focal splenic lesions are detect incidentally at US. However, we are well aware that this does not apply to other countries, where most incidental findings come from CT. Additionally, CEUS is not available worldwide and in many countries its use for non-liver applications is regarded as off label.
